# Research and Evaluation Method for Echelon Usage of Double-Life Equipment Considering Calendar Life

**Pisheng Qin \*** , **Xiaofeng Zhang, Qinglin Miao** and **Nachuan Liu**

School of Equipment Management and UAV Engineering, Air Force Engineering University, Xi'an 710051, China
\* Correspondence: qps00700@163.com

**Featured Application: Calendar life is taken as the abscissa of the double-life equipment echelon usage diagram, the controllable range of the working life is analyzed, and the calculation and evaluation methods of the optimal working life distribution are studied.**

**Abstract:** Echelon usage is an important usage and control method for double-life equipment considering calendar life, which can improve the utilization rate of equipment life and give full play to the economic benefits of the equipment. However, the life indicator considered by the most common echelon usage method is single, and most methods evaluate the echelon usage through the echelon uniformity of working life. The conclusions obtained are limited for the usage and control of double-life equipment, so it is necessary to study the life evaluation method considering calendar life. In this paper, by analyzing the life consumption law of double-life equipment, the recursive method is used to calculate the controllable life range in the process of echelon usage. The evaluation method of working life distribution considering calendar life is proposed. The equipment life reserve is evaluated according to the unit task quantity of the equipment. The results show that the new method can quickly analyze the equipment beyond the usage and control range, evaluate the distribution of working life more accurately, reflect the shortage of equipment reserve life, and has guiding significance for the usage and control of equipment.

**Keywords:** double-life equipment; calendar life; echelon usage; usage and control; controllable range

## 1. Introduction

The double-life equipment studied in this paper [1,2] refers to disposable equipment with calendar life, working life, or other life limitations, which will be used many times in the whole life cycle according to the different tasks undertaken, in which the calendar life decreases with the change of calendar time [3,4] (the difference in the reduction of calendar life caused by the working environment is not considered for the time being), and the working life will decrease according to the different use and consumption of the tasks undertaken. In order to give full play to the economic benefits of the equipment [5], it is necessary to control the use of the equipment so that the working life and calendar life are consumed and used at the same time. Due to the long life cycle of double-life equipment, the waste of life often does not occur immediately, and the problem is not easy to detect. The reasonable use control of equipment and the orderly use of reasonable life distribution are the premise to avoid the problem of life waste. For example, too little use causes the problem of working life waste, too much use causes the problem of working life reaching the specified life in advance, and improper use causes the problem of the working life of a large amount of equipment reaching the specified life at the same time.

The so-called working life echelon usage [6] refers to the equipment's working life in ascending or descending order. The arrangement of the working life of the equipment is in a step shape. The working life echelon usage of equipment is usually used to control the use of the equipment to ensure orderly use and avoid life waste. For the usage and control

decision of double-life equipment [7], it is more complex and difficult to control the use of equipment because of the need to take care of two life indicators at the same time. The current usage and control method has several main problems:

1.  Considering working life and calendar life indicators at the same time, and how to give consideration to both indicators;
2.  When the unit task quantity is determined, how to determine the controllable range of working life and whether there is equipment beyond the controllable range;
3.  What distribution of working life is the strongest resistance to the impact of working life caused by changes in the external environment;
4.  How to evaluate the equipment life reserve in the unit.

For example, a unit has 10 pieces of equipment. When the calendar life or working life of the equipment reaches its end, the equipment needs to be scrapped. Each task requires the use of multiple pieces of equipment. In order to avoid the problem of life waste and the simultaneous retirement of multiple pieces of equipment, the use of equipment should be controlled. As shown in Figure 1, two life echelon usages are marked in the control method of the ladder chart. The diagonal column is the working life, and the blank column is the calendar life. The column chart is arranged from small to large according to the working life, so that the column chart of the calendar life is not necessarily a ladder chart. If the unit task quantity is determined, how can the controllable range of life be determined? It is obvious from the figure that the calendar life of equipment No.3 is too much, and the calendar life of equipment No.7 is too little. The working life of equipment No.3 can be consumed in the same proportion as the calendar life by reducing the usage. The reduced working time of equipment No. 3 will be consumed by other equipment, and may not be consumed. Assume that the current equipment in Figure 1 is within the controllable range, and the unit task volume and other indicators change, resulting in the change to the controllable range. The distribution status of the equipment life distribution can still be within the controllable range. Under the premise that the unit is in the optimal use control strategy, how long will the equipment with calendar life and remaining working life occur under the condition of less total use, or how long will the equipment with calendar life and remaining working life occur under the condition of more use?

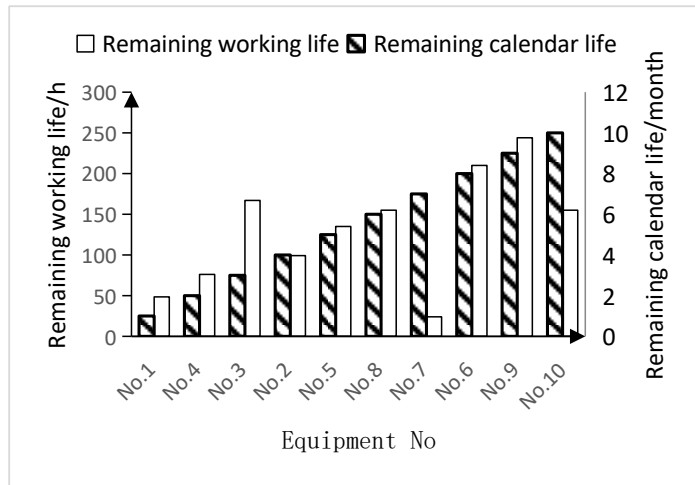

**Figure 1.** Two life echelon usage.

In order to solve the echelon usage problem of taking calendar life into account, on the basis of the control method of the ladder chart, Wang, Q. et al. [6,8] designed a control method taking calendar life into account, taking calendar life as a reference correction index of working life, and proposed a balanced life control method taking calendar life into account, but did not analyze the controllable range, and used the residual life ladder variance as an evaluation index. Zhao, J. [9] mentioned the impact of the number of pieces of equipment on the life utilization rate when studying the integrity of equipment, but did

not carry out in-depth research. Previous research has limited ability to solve the problem in this paper, so we should consider two life indicators at the same time, study the change rule of working life under different task intensities, analyze the controllable range of life, derive the optimal working life distribution, and evaluate the equipment reserve life.

## 2. Problem Description

Double-life equipment can generally improve the equipment life utilization rate by optimizing the usage and control management [10,11], improving the equipment performance [12,13] or adjusting the task [14]. As a task-undertaking unit, it is usually difficult to adjust the task or improve the equipment performance. What can effectively improve is the usage and control management ability of the unit. The echelon usage method is a common method of usage and control management, which keeps the equipment life within the specified range, ensures the equipment is in a good performance state during use [15], and reduces the risk of failure [16]. The problem is analyzed and described [17–21] by referring to the analysis of double-life equipment use process and simulation.

### 2.1. Problem Description

A unit undertaking certain use tasks has $N$ sets of double-life equipment. The initial working life of the equipment No.$i$ is $Twl_i^0$, and the initial calendar life is $Tcl_i^0$, $i \in [1, N]$. The calendar life ($Tcl_i$) will only decrease by one month with the change of calendar time, and the working life ($Twl_i$) will decrease according to the use of the unit. During the $J^0$ month period, the unit will perform $K_j$ tasks in the $j$ month, and will use $M_{j,K_j}$ equipment, consuming $U_{j,K_j}$ hours of equipment working life. The equipment No.$i$ consumes $x_{i,j}$ hours in the $j$th month. In the $j$th month, the total working life consumption is $Tyl_j$ hours, and there are:

$$Tyl_j = \sum_{i=1}^{N} x_{i,j} = \sum_{k=1}^{K_j} M_{j,k} \times U_{j,k}$$

The equipment calendar life is taken as the abscissa of the ladder diagram, and the working life is taken as the ordinate to draw the ladder diagram. After $J^0$ months of use, the following three problems can be solved:

1.  Find the controllable range of life under the current indicators; that is, analyze the relationship between calendar life and working life to keep the equipment in the controllable range and be affected by those indicators.
2.  Find the optimal distribution curve of the working life that is least affected by external indicators. That is, if N equipment is in the controllable range, after the changes of external indicators, the calendar life of the equipment that appears outside the controllable range is the largest, and the reserved time to deal with the equipment outside the controllable range is the longest. The longer the time is, the more means can be used to deal with it, which shows that the impact of external indicators is the least.
3.  Find the optimal life reserve. When the distribution of working life is least affected by external indicators, how much should the total of the working life be?

### 2.2. Problem Assumptions

To facilitate the disposal of the remaining waste of calendar life, that is, when the working life is 0 ($Twl_i = 0$), the calendar life is not 0 ($Tcl_i \neq 0$), resulting in the remaining waste of calendar life ($Tcl_i$). The equipment can be stored in the warehouse and suspended for use, that is, the equipment with a working life ($Twl_i$) approaching 0, but not equal to 0, and a calendar life is not 0 ($Tcl_i \neq 0$) can be stored in the warehouse and reused when the calendar life ($Tcl_i$) is about to expire. The scrapping standard of equipment working life is different, that is, when $Twl_i \leq T^0$, the equipment is considered to be scrapable, and the standard of equipment working life ($T^0$) is different. Therefore, the working life value can be taken to be close to 0, but not equal to 0, that is, the working life $Twl_i \approx 0$, and the

equipment with a calendar life $Tcl_i \neq 0$ is regarded as normal. When the equipment is used again, the problem that no equipment is available is defined as that there is not enough equipment to use, and the problem of calendar life waste is converted into that there is enough equipment to use.

- Assumption 1: The number of tasks undertaken by the unit each month is fixed, that is, the working life of the consuming equipment is a fixed value equal to its average monthly service time in the future, that is:

$$Ty = \sum_{j=1}^{J} \frac{Tyl_j}{J}$$

- Assumption 2: When a piece of equipment reaches its service life, a new one will be added immediately and once a month;
- Assumption 3: The quantity of equipment required for each time is fixed. $M_{j,K_j} = M$;
- Assumption 4: Equipment scrapping in advance caused by other indicators such as failure is not considered.

## 3. Analysis and Solution

The echelon usage of equipment is a multi-objective decision-making problem. Liu, Q. et al. [22] elaborated the evaluation indicators of multiple objectives, especially the quantitative indicators of life echelon uniformity for working life. However, they did not propose the optimal echelon interval determination of echelon uniformity. According to the use control ability and the comprehensive utilization rate, Liu, J. [23] and Zhang, H. [24] respectively put forward the calculation method of the optimal step interval to evaluate the life distribution more accurately. Dual-life equipment needs to consider two life indicators. Zhang, Q. et al.'s [25] treatment method is to use the analytic hierarchy process to weight the two life indicators. Li, L. et al. [26] deal with the relationship between the two life indicators by calculating the proportion of the optimal double life based on historical data. Cheng, X. et al. [27,28] used a BP neural network to process the weights of various indicators and ranked the equipment use. Miao, Q.-L. et al. [29] proposed to arrange the equipment according to the size of the calendar life, and then use the echelon uniformity to evaluate the working life, but this method is only suitable for the specific life distribution. This paper considers two life indicators at the same time, analyzes the law of life change under different circumstances, proposes the calculation method of the controllable range of equipment under different working life and numbers used, and finally calculates the optimal working life distribution and the sum of the optimal working life when the external environment changes.

### 3.1. An Extreme Special State: $M = N$

The number of double-life equipment items in a unit is $N$. The equipment number is $i, i \in [1, N]$ (the maximum number of the current number is incremented). The remaining calendar life is $Tcl_i$. The working life and the calendar life of all equipment are shown in Figure 2, respectively. The unit undertakes a fixed task every month, and the working life monthly usage is $Ty$. One new piece of equipment will be added every month. The calendar life of the new equipment is $N$ months, and the remaining working life is $Ty$. The number of units used each time is $M = N$.

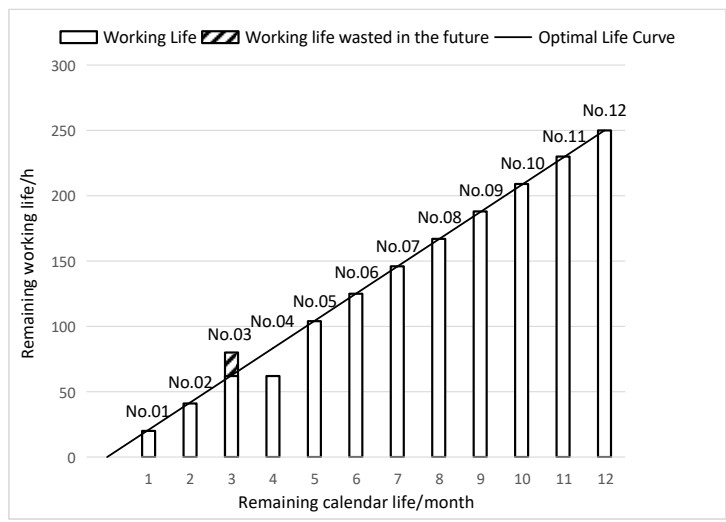

**Figure 2.** Echelon usage diagram with $M = N$.

From the number of units used, $M = N$, it can be seen that the working life consumption of each equipment item per month is $Ty \div M$.

To meet the requirement of no waste of working life, the relationship between the working life and calendar life of each equipment item will meet the following requirements:

$$Twl_i \leq Tcl_i \times \frac{Ty}{M}$$

To meet the requirement of sufficient equipment, the relationship between the working life and calendar life of each equipment item will meet the following requirements:

$$Twl_i \geq Tcl_i \times \frac{Ty}{M}$$

At the same time, there is no waste of working life and there are enough pieces of equipment to use. The relationship between the working life and calendar life of each set will meet the following requirements:

$$\begin{cases} Twl_i \leq Tcl_i \times \frac{Ty}{M} \\ Twl_i \geq Tcl_i \times \frac{Ty}{M} \end{cases} \Rightarrow Twl_i = Tcl_i \times \frac{Ty}{M}$$

Therefore, the optimal distribution curve of the extreme special state can be obtained:

$$Twl = \frac{Ty}{M} \times Tcl$$

Since all equipment items are used, there is only one choice, that is, all equipment is used, so the controllable range is also the optimal distribution curve. The relationship between the working life of all of the equipment and the calendar life must be within the controllable range to ensure that there is no waste of working life or calendar life. As shown in Figure 2, if the diagonal part of equipment No.03 is used according to the current specific rules, the working life of this part must not be used up, which is bound to cause a waste of working life. If equipment No.04 is used according to the current specific rules, its working life will be consumed in the fourth month, resulting in insufficient equipment.

### 3.2. When the Number of Units Used Each Time Is Less Than Total: $M < N$

The number of double-life equipment items in a unit is $N$. The equipment number is $i, i \in [1, N]$ (the maximum number of the current number is incremented). The remaining calendar life is $Tcl_i$. The working life and the calendar life of all equipment items are shown

in Figure 3, respectively. The unit undertakes a fixed task every month, and the working life monthly usage is $Ty$. One new equipment item will be added every month. The calendar life of the new equipment is $N$ months, and the remaining working life is $Ty$. The number of units used each time is $M < N$.

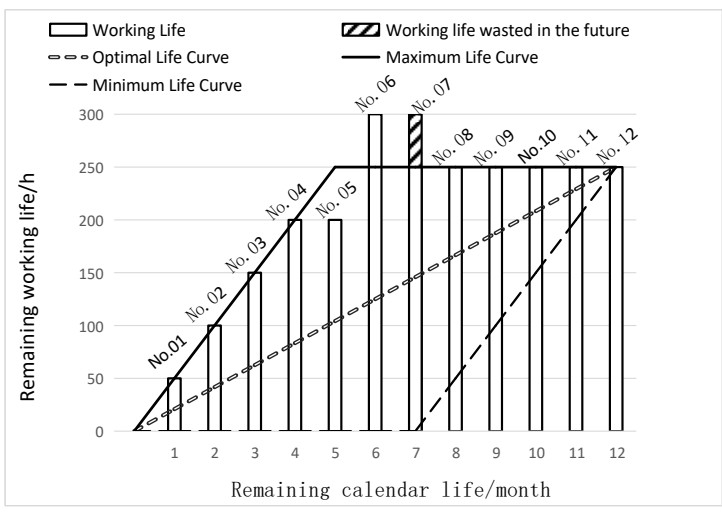

**Figure 3.** Echelon usage diagram with $M < N$.

### 3.2.1. Maximum Working Life Controllable Range

Based on the number of units used, $M < N$, the consumption of each working life is uncertain, but it can be seen that the maximum consumption of a single unit of equipment per month is $Ty \div M$. No matter how much a single piece of equipment is used, it cannot exceed this amount, so the working life ($Twl_i$) and calendar life ($Tcl_i$) should meet the following conditions:

$$\frac{Twl_i}{Tcl_i} \leq \frac{Ty}{M}, i \in [1, N], M \leq N$$

Under the condition that the above formula is satisfied, there is still a situation that life waste cannot be avoided through usage and control. As shown in Figure 3, the working life of the diagonal part of equipment No.07 will be wasted.

Assuming that the working life of the first $i$ equipment is within the maximum life curve, the optimal decision can be found through usage and control, and the optimal decision can be found. The optimal decision in the first month is $f(1)$, and the optimal decision in the $Tcl_i$ month is $f(Tcl_i)$. After $Tcl_i$ months, the working life of the $i$ equipment and the previous equipment is consumed at the end of the calendar life, and then the maximum working life of the first $i$ equipment in $Tcl_i$ months for the task consumption is:

$$Max(\sum_{j=1}^{i} Twl_j) = \sum_{j=1}^{Q} Ty + \sum_{j=P}^{Tcl_i} \left(\frac{Tcl_i - j + 1}{M} Ty\right)$$
$$Q = \text{ReLU}(Tcl_i - M)$$
$$P = \text{ReLU}(Tcl_i - M + 1)$$

The maximum working life of the equipment No.$i$ is:

$$MaxTwl_i = Max(\sum_{j=1}^{i} Twl_j) - \sum_{j=1}^{i-1} Twl_j$$

The controllable range of the equipment No.$i$'s working life is:

$$0 \leq Twl_i \leq \min(MaxTwl_i, Tcl_i \frac{Ty}{M}) \tag{1}$$

The maximum life curve is drawn according to the maximum life value of each equipment item, as shown in Figure 3.

When the working life of equipment No.$i - 1$ is within the controllable range, and equipment $i$ meets the above formula at the same time, the working life of the equipment $i$ is within the controllable range. Using the idea of dynamic programming, the constraint conditions of the controllable range of working life are analyzed and solved. That is to say, to determine the controllable range of the working life of the $i$ equipment, we should first find out whether the working life of equipment No.$i - 1$ is within the controllable range, and by analogy, we can determine whether the working life of the subsequent equipment is within the controllable range from the working life of the first piece of equipment.

### 3.2.2. Minimum Working Life Controllable Range

From the number of units used, $M < N$, it can be seen that the maximum consumption of the working life of a single item of equipment is $Ty \div M$ each month, so the working life of the equipment is not completely effective and only part of the working life can be consumed. If condition 3.2.1 is met, the effective working life of the equipment $i$ in the $k_i$ month of use is $VTwl_i = Min(k_i \times \frac{Ty}{M}, Twl_i)$. Since the starting time of the equipment is the same, the effective working life of the equipment in the $J^1$ month is $AVTwl_{J^1}$:

$$AVTwl_{J^1} = \sum_{i=1}^{N} VTwl_i = \sum_{i=1}^{N} Min(L \times \frac{Ty}{M}, Twl_i)$$

$$L = Min(J^1, Tcl_i)$$

Since there is a supplement of new equipment during the service period, that is, the new equipment can also be used to undertake the consumption of tasks, $N'$ new equipment is added, the equipment No.$i$ is added to the unit at $j_i$ month ($j_i < J^1$), and the equipment $i$ is put into use in $k_i = J^1 - j_i$ month, and then the formula of the effective working life provided by the new equipment is:

$$NVTwl_{J^1} = \sum_{i=N+1}^{N+N'} Min(k_i \times \frac{Ty}{M}, Twl_i)$$

In order to have enough equipment for use in the $J^1$ month, the following conditions must be met:

$$AVTwl_{J^1} + NVTwl_{J^1} \geq J^1 \times Ty \tag{2}$$

Assumption 2 in this paper shows that new equipment is supplemented every month. When enough equipment is used in the $J^1 = M$ month, the supplemented $M - 1$ new equipment and the newly supplemented equipment in the next month can meet the task requirements in the $J^1 = M + 1$ month.

The fixed formula of effective working life provided by the new equipment is:

$$NVTwl_{J^1} = \sum_{i=1}^{M-1} i \times \frac{Ty}{M}$$

The minimum working life sum of the first $N$ equipment can be calculated as:

$$Min(\sum_{i=1}^{N} Twl_i) = M \times Ty - \sum_{i=1}^{M-1} i \times \frac{Ty}{M}$$

Draw a straight line through the point $(Tcl_N, Ty)$ with the slope of $\frac{Ty}{M}$. The area of the right triangle formed with the X axis is equal to the minimum value of the working life sum of the first $N$ set of equipment. This straight line is defined as the minimum life curve, as shown in Figure 3.

### 3.2.3. Calculate the Optimal Distribution of Working Life

There are many kinds of equipment life distributions that meet the two conditions of no working life waste and enough equipment use at the same time. However, under the same service conditions, the changes of the sum of the working life of all of the equipment are the same, and the optimal distribution curve of the working life is in these conditions.

By analyzing how the maximum life curve of the working life is affected by the monthly usage ($Ty$) and the number of units used ($M$), it can be seen that when the monthly usage ($Ty$) decreases or the number of units used ($M$) increases, the controllable range becomes smaller, which will cause the equipment to move beyond the controllable range. The controllable range changes are shown in Figure 4.

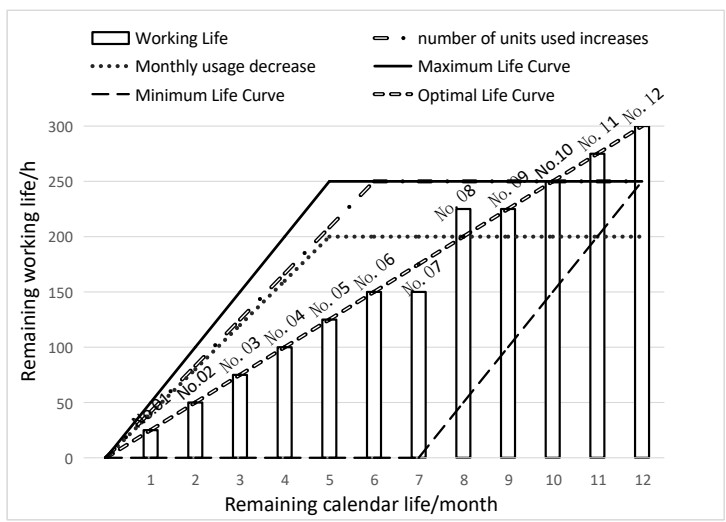

**Figure 4.** Change in monthly usage or the number of units used.

The maximum sum of the working life chart affected by monthly usage and number of units can be drawn as in Figure 4.

By observing Figure 5, it can be seen that under the same working life and conditions, the sum of working life at any time should be lower than the value of $M = N$ curve, and number of units used $M = N$ also includes another requirement, $\frac{Twl_i}{Tcl_i} \geq \frac{Twl_{i-1}}{Tcl_{i-1}}$. Therefore, the minimum working life affected by external indicators will meet:

$$\frac{Twl_i}{Tcl_i} \geq \frac{Twl_{i-1}}{Tcl_{i-1}} \tag{3}$$

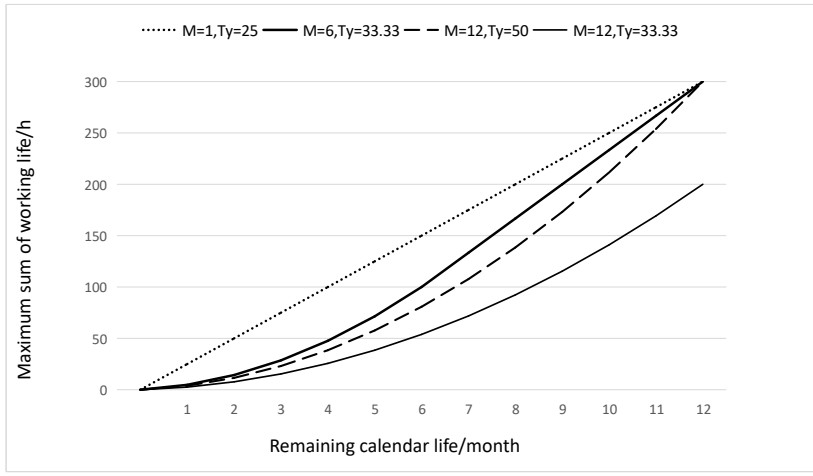

**Figure 5.** Maximum sum of working life.

Enough equipment should be used in the month to meet the following conditions:

$$AVTwl_{J^1} + NVTwl_{J^1} \geq J^1 \times Ty$$

It can be seen that under the same working life and conditions, the greater the number of units used, the more limited they become. So, set $M_0 = N$, $J^1 = N$. $NVTwl_{J^1}$ are taken as fixed values, and the following formula is obtained:

$$\begin{cases} AVTwl_{J^1} = \sum\limits_{i=1}^{N} Min(Tcl_i \times \frac{Ty}{N}, Twl_i) \\ \sum\limits_{i=1}^{N} Tcl_i \times \frac{Ty}{N} = N \times Ty - NVTwl_N \end{cases}$$

Further, the following formula can be obtained:

$$Tcl_i \times \frac{Ty}{N} \leq Twl_i$$

Different values can be taken for the monthly usage, such as $Ty_1 \leq Ty_2 \leq Ty_{\max}$. Set $Ty = Ty_{\max}$:

$$Tcl_i \times \frac{Ty_{\max}}{N} = Twl_i$$

Further, we can obtain the formula:

$$\frac{Twl_i}{Tcl_i} = \frac{Twl_{i-1}}{Tcl_{i-1}} = \frac{Ty_{\max}}{N} \tag{4}$$

So, the result of meeting the two formulas (3) and (4) under the same sum of working life is:

$$\frac{Twl_i}{Tcl_i} = \frac{Twl_{i-1}}{Tcl_{i-1}}$$

Thus, the slope of the optimal working life distribution curve, which has the same sum of working life, can be obtained as follows:

$$k = \frac{\sum\limits_{i=1}^{N} Twl_i}{\sum\limits_{i=1}^{N} Tcl_i} \tag{5}$$

3.2.4. Calculate the Optimal Value of Total Working Life

As the controllable range is affected by the working life monthly usage ($Ty$) and the number of units used each time ($M$), the optimal value of the total working life ($D$) should be the average of the maximum and minimum:

$$D = Ty \times \frac{(Tcl_N + 1)}{2}$$

*3.3. More Common Status*

Cancel the initial three assumptions to make the problem more general, and the working life monthly usage is not fixed. The time of new equipment replenishment is uncertain, and the number of units used each time is not fixed.

The calendar life interval of adjacent equipment caused by the uncertain replenishment time of the new equipment is not equal to 1 month. However, it can be seen from formulas (1) and (2) that the calendar life interval value of the equipment is independent of the calculation of the controllable range.

According to Formula (5), the optimal distribution curve of the working life is only related to the sum of working life and calendar life. The canceled assumption does not affect the optimal distribution curve of the working life.

Therefore, the calculation method is still valid after the assumption is canceled.

## 4. Evaluation Method

In view of the problems that exist in the echelon usage of double-life equipment, it is necessary to evaluate the current life status of the equipment in order to provide guidance for the usage and control of the equipment.

### 4.1. Previous Evaluation Indicators

In previous studies [2,6,10,15,22], the distribution of working life is mainly evaluated by the echelon uniformity $Q_1$ and the life reserve $Q_2$. The evaluation calculation formula is as follows.

$$Q_1 = \left(1 - \sqrt{\frac{\sum\limits_{i=1}^{N-1}(Twl_{i+1}-Twl_i-\Delta)^2+(Twl_1+Twl_h-Twl_N-\Delta)^2}{(N-1)\Delta^2+(Twl_h-\Delta)^2}}\right) \times 100\%$$

$$\Delta = \frac{Twl_h}{N}, Twl_h = \text{MAX}(Twl_i), i \in [1, N]$$

$$Q_2 = \frac{\sum Twl_i}{\sum Twl_i^0} \times 100\%$$

The closer $Q_1$ is to 100%, the better; $Q_2 > 40\%$ is better, and the bigger the better.

### 4.2. Evaluation Indicators in This Paper

In this paper, the optimal distribution curve of working life is used to describe the optimal working life distribution, and the echelon difference $Q_3$ is introduced to evaluate the working life distribution, so that the working life distribution is consistent with the optimal distribution curve. The smaller the echelon difference, the better. Since the future time has little impact on the current time, the working life should be corrected according to the calendar life, and $\omega$ is the impact coefficient. The formula $Q_3$ is:

$$Q_3 = \frac{1}{N}\sum_{i=1}^{N}\left(\frac{Twl_i - k \times Tcl_i}{1 + \omega \times Tcl_i}\right)^2 \tag{6}$$

Because the future use is uncertain, the estimated working life monthly usage ($Ty'$) is used to estimate the optimal value of the total working life, and the reserve priority $Q_4$ is introduced for evaluation. The closer to 100%, the better. The formula $Q_4$ is:

$$Q_4 = \frac{\sum\limits_{i=1}^{N} Twl_i}{Ty' \times \frac{(Tcl_N+1)}{2}} \times 100\% \tag{7}$$

## 5. Examples and Results

Take a double-life airborne item as an example. The units equipped with this type of equipment are Unit 1 to Unit 3. The estimated working life monthly usage ($Ty'$) is 300 h, 150 h, and 150 h, respectively. The estimated number of units used each time is 7 ($M = 7$). The equipment working life ($Twl$) and calendar life ($Tcl$) are shown in Table 1. The working life ($Twl$) is measured in hours, the calendar life ($Tcl$) is measured in months, and the initial working life value is 300 h, with an influence coefficient $\omega = 0.1$.

**Table 1.** Equipment life information.

| No. | Unit 1 | | Unit 2 | | Unit 3 | |
|---|---|---|---|---|---|---|
| | *Twl* | *Tcl* | *Twl* | *Tcl* | *Twl* | *Tcl* |
| 1 | 25 | 2 | 25 | 1 | 12.5 | 1 |
| 2 | 50 | 4 | 50 | 2 | 25 | 2 |
| 3 | 95 | 6 | 75 | 6 | 75 | 6 |
| 4 | 100 | 8 | 100 | 7 | 87.5 | 7 |
| 5 | 125 | 10 | 125 | 8 | 100 | 8 |
| 6 | 150 | 12 | 150 | 9 | 112.5 | 9 |
| 7 | 175 | 14 | 175 | 13 | 162.5 | 13 |
| 8 | 200 | 16 | 200 | 20 | 250 | 20 |
| 9 | 225 | 18 | 225 | 21 | 262.5 | 21 |
| 10 | 250 | 20 | 270 | 22 | 295 | 22 |
| 11 | 275 | 22 | 275 | 23 | 287.5 | 23 |
| 12 | 300 | 24 | 300 | 24 | 300 | 24 |

The three units were evaluated according to the evaluation method. Calculate the slope of the optimal distribution curve ($k$), echelon uniformity ($Q_1$), life reserve ($Q_2$), echelon difference ($Q_3$), and reserve priority ($Q_4$) of each unit, as shown in Table 2.

**Table 2.** Calculation results.

| Item | Unit 1 | Unit 2 | Unit 3 |
|---|---|---|---|
| slope of optimal distribution curve ($k$) | 12.63 | 12.63 | 12.63 |
| echelon uniformity ($Q_1$) | 90.15% | 90.15% | 70.45% |
| life reserve ($Q_2$) | 54.72% | 54.72% | 54.72% |
| echelon difference ($Q_3$) | 12.50 | 138.98 | 2.81 |
| reserve priority ($Q_4$) | 52.53% | 105.07% | 105.07% |

By analyzing the calculation results, it can be seen that the conclusion obtained by echelon difference ($Q_3$) is contrary to the conclusion of the echelon uniformity ($Q_1$). The controllable range recursive calculation method (1) (2) was used to check the equipment life of units 1–3, and the results are shown in Figure 6. The equipment in Unit 1 and Unit 3 is within the controllable range. Equipment No.1 and No.2 in Unit 2 is outside the controllable range, and the working life will be wasted in the future. The conclusion obtained using the echelon difference ($Q_3$) is more accurate.

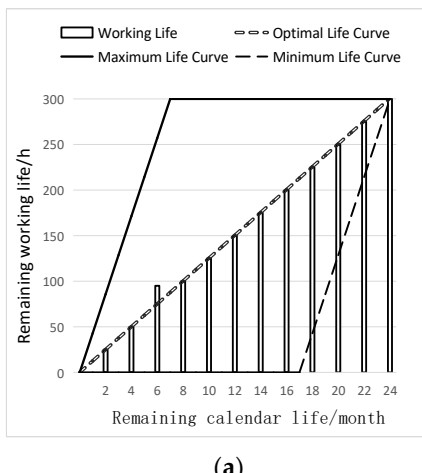

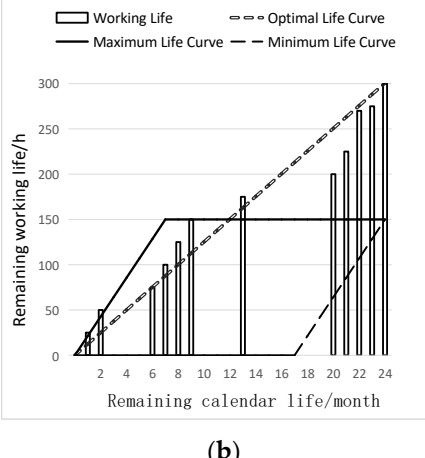

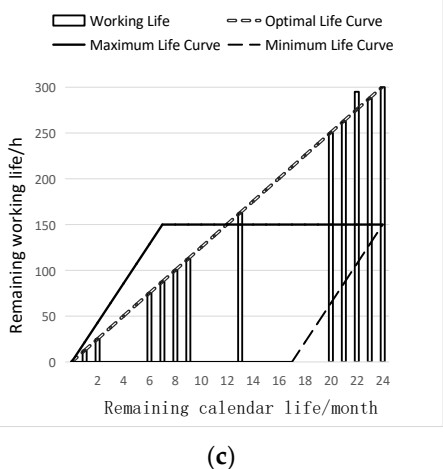

(a)  (b)  (c)

**Figure 6.** Check of the controllable range of Units 1–3. (**a**) Check of Unit 1; (**b**) check of Unit 2; (**c**) check of Unit 3.

The evaluation results of echelon uniformity for Unit 1 and Unit 2 are the same, and the difference in the life distribution cannot be distinguished. By evaluating Unit 1 and Unit 3 with the echelon difference, the difference of the optimal order of different life distribution can be clearly distinguished.

Life reserve ($Q_2$) evaluation does not distinguish the evaluation under the condition of the same sum of working life, and has limited guidance on the degree of reserve life shortage and future equipment replenishment. According to the calculation result of the reserve priority ($Q_4$), it can be determined that the reserve priority ($Q_4$) of Unit 1's equipment working life is lower than that of Unit 2 and Unit 3. That is, if new equipment is not supplemented within 24 months, Unit 1 will experience an equipment shortage earlier than Unit 2 or Unit 3. Therefore, Unit 1 should be given priority when supplementing new equipment.

## 6. Discussion

From the results obtained, it can be seen that under the condition of uniform calendar life, the echelon difference is consistent with the evaluation obtained using the echelon uniformity calculation results, and both believe that the working life distribution of Unit 1 is very good. However, when the calendar life is nonuniform, the results are quite different. The reason for the difference is that the echelon uniformity method does not consider the difference caused by the uneven calendar life. The echelon difference method proposed in this paper eliminates this difference and makes the results more accurate. The validity of the echelon difference evaluation can be verified by the recursive calculation method of the controllable range.

Life reserve ($Q_2$) evaluation does not distinguish the evaluation under the condition of the same sum of working life, because the impact of unit task consumption is not considered. According to the calculation method of the optimal life distribution, the evaluation method of reserve priority proposed in this paper can take into account the unit task consumption and is more meaningful for the evaluation of the same sum of working life.

In this paper, the optimal life distribution is calculated according to the estimated future state, but the change in the future state of each unit in the process of use is more complex, so the accuracy of the future state prediction is very important. In addition to double-life equipment, there is also similar multi-life equipment, which considers more life indicators. The controllable range and evaluation method of more life indicators can be further studied on the basis of this paper.

## 7. Conclusions

Aiming to solve the problem of using the control process of double-life disposable equipment considering calendar life, the analysis and solution method proposed in this paper can take into account calendar life and working life. The controllable range of working life can be effectively analyzed and the equipment beyond the controllable range can be found according to the working life monthly usage and the number of units used each time. According to the derived optimal working life distribution, the equipment working life distribution can be evaluated effectively by echelon difference. The degree of shortage of each unit's demand for new equipment can be evaluated using the reserve superiority. It has guiding value for the echelon usage of double-life equipment.

**Author Contributions:** Algorithm, P.Q.; validation, P.Q., X.Z. and Q.M.; writing—original draft preparation, P.Q.; writing—review and editing, P.Q.; formal analysis, P.Q. and N.L.; investigation, P.Q. and Q.M.; resources, P.Q. and X.Z.; data curation, Q.M. All authors have read and agreed to the published version of the manuscript.

**Funding:** This research received no external funding.

**Institutional Review Board Statement:** Not applicable.

**Informed Consent Statement:** Not applicable.

**Data Availability Statement:** Not applicable.

**Conflicts of Interest:** The authors declare no conflict of interest.

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
