# Peer review of "Research and Evaluation Method for Echelon Usage of Double-Life Equipment Considering Calendar Life"

_applsci, doi:10.3390/app13031655_

Round 1

Reviewer 1 Report

Even if the paper is well organised, several comments can be developed in the way to improve paper.
1- In section 1, it will be pertinent to give a concrete example on a equipment  with regards to Echelon Usage, double life, calendar life ... It will allows to underline to main pratical challenges.
2- Section 1 does not higlight the scientific issues extracted from the state of the art you done (scientific issues that you want to attack  y your contributions). Indeed the SoA led to a list of references but without clearly identify what are the scientific problems not already solved today in consistence with the paper topic. Moreover, your conclusion is not really related to the items you identified in the last paragraph of section 1.

3- Several problems of formatting are present in the paper, as in p3 and p4

4 - Could you precise, the main scientific novelty of your contributions in section 3.

5- In section, you proposed Equipment life information. But what are concretely these equipments? Could you detail more the current situation (in terms of what you want to improve) which is not optimal (as you used it for testing your approach)?  Discussion is not pertinent as you did not compare the results you obtained with regards to current values?

6- Sections 5 and 6 are not eough detailly to prove the interest of your approach.

Author Response

Thanks for your comments.

Reviewer 2 Report

Paper lacks novelity

Conclusions should be more defined

Research work should be shown with more figures

Author Response

Thanks for your comments.

Reviewer 3 Report

Authors are well defined the article but there are few questions to the authors

1.  Abstract to be revised towards the problem definition, methodology, novelty, results and conclusion of the study.

2. Introduction section found that the lack of  recent literatures, so authors should study  the extensive  recent literature survey and update it.

Author Response

Thanks for your comments

Round 2

Reviewer 1 Report

The revised version bring more added value to the initial paper.

Author Response

Dear professor:

Thank you very much for your comments.

The following modifications have been made this time:

1.Some modifications have been made to English language and style.

2.Some details have been added to the description and content of the calculation method.

3.Some updates have been made to the presentation of the chart content.

And thank you again for your comments.

Reviewer 2 Report

Authors have done significant work

Author Response

(The authors gave the same response as above.)
